# GLEN: Generative Retrieval via Lexical Index Learning

**Sunkyung Lee**[*], **Minjin Choi**[*], **Jongwuk Lee**[†]
Sungkyunkwan University, Republic of Korea
{sk1027, zxcvxd, jongwuklee}@skku.edu

## Abstract

Generative retrieval shed light on a new paradigm of document retrieval, aiming to directly generate the identifier of a relevant document for a query. While it takes advantage of bypassing the construction of auxiliary index structures, existing studies face two significant challenges: (i) the discrepancy between the knowledge of pre-trained language models and identifiers and (ii) the gap between training and inference that poses difficulty in learning to rank. To overcome these challenges, we propose a novel generative retrieval method, namely *Generative retrieval via LExical iNdex learning (GLEN)*. For training, GLEN effectively exploits a dynamic lexical identifier using a *two-phase index learning strategy*, enabling it to learn meaningful lexical identifiers and relevance signals between queries and documents. For inference, GLEN utilizes *collision-free inference*, using identifier weights to rank documents without additional overhead. Experimental results prove that GLEN achieves state-of-the-art or competitive performance against existing generative retrieval methods on various benchmark datasets, *e.g.*, NQ320k, MS MARCO, and BEIR. The code is available at https://github.com/skleee/GLEN.

## 1 Introduction

Generative retrieval has emerged as an innovative approach to document retrieval (Metzler et al., 2021). Unlike conventional retrieval methods following the "index-retrieve-then-rank" pipeline, it unifies an entire search process. Specifically, it directly generates the identifier of a relevant document for a given query. By formulating the entire search process as a sequence-to-sequence problem, it bypasses an auxiliary index structure and can be optimized through end-to-end learning.

Despite these benefits, generative retrieval faces major challenges in how to define and train document identifiers. As depicted in Table 1, existing studies are categorized into two pillars: *identifier types* and *identifier learning strategies*.

**Identifier types.** Some canonical works employ *numeric* identifiers for document representation, *e.g.*, hierarchical clustering using document representations (Tay et al., 2022; Wang et al., 2022) and product quantization (Zhou et al., 2022). However, numeric identifiers struggle to fully exploit the knowledge of pre-trained language models (PLMs) due to a semantic discrepancy between natural language and numeric identifiers. Other studies pre-define *lexical* identifiers using titles (Lee et al., 2023) or URLs (Zhou et al., 2022; Ren et al., 2023). Although they can narrow the semantic gap between PLM knowledge and identifiers, such information may be inadequate for representing documents and does not exist depending on the dataset.

**Identifier learning strategies.** Depending on the strategy of training identifiers, we refer to an identifier as being *static* if it does not change during training. Meanwhile, when an identifier evolves during training, we refer to it as being *dynamic*. Static identifiers may lead to a performance bottleneck in generalizing for unseen documents during training. To overcome this limitation, Sun et al. (2023) proposed a method to dynamically learn numeric identifiers. However, it is still non-trivial to learn appropriate identifiers due to the task discrepancy between training and inference; the models focus on generating the identifier during training, but they need to rank documents during inference.

To this end, we introduce a new generative retrieval method, namely *Generative retrieval via LExical iNdex learning (GLEN)* using the *dynamic lexical identifier* in a right-bottom cell in Table 1. The key novelty of GLEN is (i) to define lexical identifiers from documents using the knowledge of PLMs, (ii) to learn them from relevance between queries and documents, and (iii) to effectively rank documents with the same identifier for inference.

---
[*]Equal contribution
[†]Corresponding author

**Training of GLEN.** It utilizes a *two-phase index learning strategy* to define lexical identifiers and to learn them dynamically. First, in the *keyword-based ID assignment* phase, GLEN defines identifiers from documents and learns them. To alleviate the discrepancy between the knowledge of PLMs and the semantics of identifiers, we depict identifiers in the pre-trained vocabulary space leveraging self-supervised signals by extracting key terms from documents. The model can learn how to map its knowledge to the unique nature of identifiers.

Then, the *ranking-based ID refinement* phase is used to effectively learn dynamic identifiers. We directly incorporate query-document relevance in learning through the elaborate design of two loss functions. Specifically, GLEN explicitly learns query-document relevance using pairwise ranking loss to capture the ranking relationships and point-wise retrieval loss to learn the relationship between a query and a relevant document. Thus, GLEN can generate identifiers that better encapsulate the subtle semantics of the query-document relationship.

**Inference of GLEN.** It employs *collision-free inference* using an identifier weight to deal with the document identifier collision problem, *i.e.*, the same identifier can be assigned to multiple documents if they are semantically similar. A simple solution is to force a different identifier for those documents. However, it can make the identifier too long or potentially interfere with the semantic learning of the identifier. Instead of enforcing uniqueness during training, we leverage the document identifier logits during inference to rank the collided documents. Notably, this simple-yet-effective solution avoids high computational costs by using the generation logit as the weight.

In summary, our key contributions are as follows. (i) We propose GLEN, which learns lexical identifiers in a dynamic fashion. To our knowledge, it is the first generative retrieval method using learning-based lexical identifiers. (ii) We devise a two-phase index learning strategy with keyword-based ID assignment and ranking-based ID refinement to generate identifiers reflecting query-document relevance. (iii) We present a collision-free inference via ranking using identifier weight while effectively preserving identifier semantics. (iv) We evaluate the effectiveness of GLEN on three benchmark datasets: Natural Questions (Kwiatkowski et al., 2019), MS MARCO Passage Ranking (Nguyen et al., 2016), and BEIR (Thakur et al., 2021).

| | Numeric | Lexical |
|---|---|---|
| Static | DSI (2022), Ultron-PQ (2022), NCI (2022), DSI-QG (2023) | GENRE (2021), SEAL (2022), CorpusBrain (2022), Ultron-URL (2022), NpDecoding (2023), TOME (2023) |
| Dynamic | GENRET (2023) | **GLEN (Ours)** |

Table 1: Category of existing generative retrieval models based on (i) identifier types and (ii) identifier learning strategies. GLEN introduces a dynamic lexical identifier.

## 2 Related Work

### 2.1 Document Retrieval

Document retrieval aims to seek relevant documents to the user query from a large document corpus. Most existing methods have followed the "index-retrieve-then-rank" pipeline. Traditional sparse retrieval methods (Robertson and Walker, 1994; Formal et al., 2021; Choi et al., 2022) rely on the inverted index utilizing term matching signals. On the other hand, dense retrieval methods (Karpukhin et al., 2020; Xiong et al., 2021; Khattab and Zaharia, 2020) calculate the vector similarity of dense representations via an approximate nearest neighbor index. Although dense retrieval has shown a remarkable performance, the model cannot be optimized end-to-end and has a drawback in the cost of the external index structure.

### 2.2 Generative Retrieval

Apart from traditional retrieval, generative retrieval uses only a unified model (Metzler et al., 2021) that directly generates an identifier of a relevant document for a given query. As shown in Table 1, we categorize the existing methods according to how they define and train identifiers.

**Numeric identifier.** Tay et al. (2022) proposed the differentiable search index (DSI), firstly demonstrating that document retrieval can be accomplished with a single model. It assigns numeric identifiers in various ways, *e.g.*, atomic, naive, and semantic. Especially, semantic identifiers are obtained by hierarchical $k$-means clustering over document representations to capture the document semantics. Wang et al. (2022) and Zhuang et al. (2023) additionally introduced the prefix-aware weight-adaptive decoder and data augmentation using query generation, respectively. To improve the semantic deficiency of numeric identifiers, Zhou et al. (2022) employed product quantization. Most

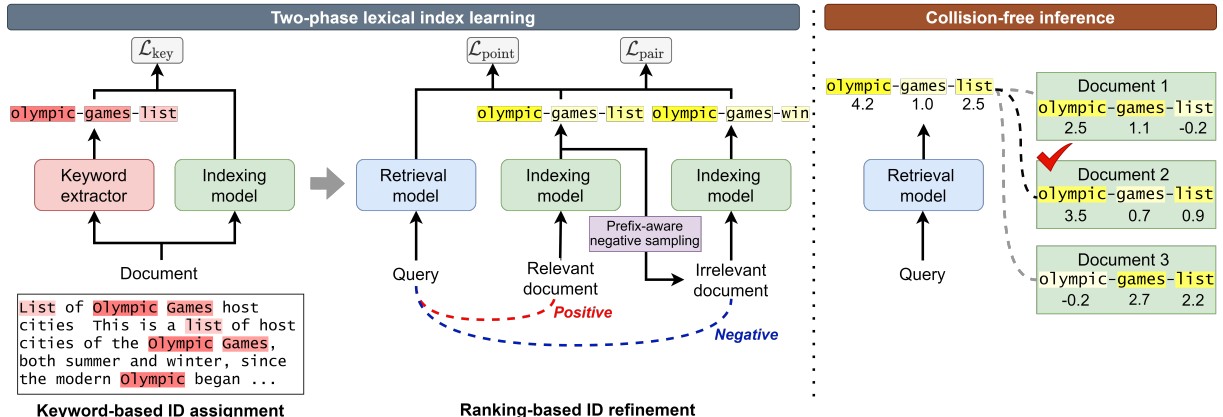

Figure 1: Overview of training and inference for GLEN. For training, the keyword-based ID assignment phase is performed, which learns identifiers via self-supervised signals, followed by the ranking-based ID refinement phase to learn identifiers dynamically. For inference, GLEN generates identifiers for a query, and the documents are ranked with the logits when the collision occurs. The number below the identifier token indicates the logit for each token.

recently, Sun et al. (2023) proposed an identifier learning framework to overcome the limitations of static identifiers. However, numeric identifiers inherently suffer from the difficulties of leveraging the knowledge of PLM due to a gap between natural language and numeric values.

**Lexical identifier.** Bevilacqua et al. (2022) proposed a method to consider n-grams in a document as identifiers using the FM-Index structure. Zhou et al. (2022) and Ren et al. (2023) utilized URLs as a document identifier, while Chen et al. (2022), Lee et al. (2023), and Cao et al. (2021) defined a title as an identifier. They can leverage the knowledge of PLMs to decode identifiers, enjoying the benefit of pre-trained vocabulary space. However, external information such as URLs and titles may not exist depending on the datasets and may not adequately represent the document. To overcome these limitations, we define lexical identifiers by extracting keywords from documents and dynamically refine them by directly optimizing query-document relevance.

## 3 Proposed Method

In this section, we formulate the generative document retrieval task (Section 3.1) and present GLEN (Section 3.2), as depicted in Figure 1. To tackle the challenges of identifier design and training strategy, GLEN adopts a *two-phase lexical index learning* (Section 3.3). For inference, we devise a *collision-free inference* using identifier logits (Section 3.4). While maintaining its simplicity, GLEN handles identifier collisions where semantically similar documents share the same lexical identifier.

### 3.1 Task Formulation

Generative retrieval aims to autoregressively generate the identifier of the relevant document for a given query. Specifically, it involves computing the probability $P(z|q)$ of generating a document identifier $z$ for the query $q$.

$$P(z|q) = \prod_{t=1}^{n} P(z_t|q, z_{<t}), \qquad (1)$$

where $n$ is the number of tokens in the identifier.

To address the key challenges of generative retrieval: (i) how to define identifiers and (ii) how to train query-document relevance, we propose a *dynamic lexical identifier* by defining it using keywords and refining it through relevance.

### 3.2 Model Architecture

We propose a novel generative retrieval method, *Generative retrieval via **LE**xical i**N**dex learning (GLEN)*. Specifically, it consists of two components: (i) An *indexing model* takes a document $d$ as input to generate a document identifier $z$, and (ii) a *retrieval model* takes a query $q$ as input to generate the identifier of a relevant document.

We describe the process of deriving identifier $z$ from document $d$ using the indexing model, and the retrieval model can proceed in the same way. Both models are initialized with the pre-trained language model with the Transformer architecture (Vaswani et al., 2017) and share parameters. For the indexing model, a document representation is defined as follows.

$$\mathbf{d}_t = \text{Dec}(\text{Enc}(d), \mathbf{d}_{<t}),$$
$$z_t = \underset{j \in \{1,...,|V|\}}{\arg\max} (\mathbf{d}_t \cdot \mathbf{e}_j), \qquad (2)$$

where $\mathbf{d}_t \in \mathbb{R}^m$ is the final hidden representation of the decoder at time $t$. An embedding vector $\mathbf{e}_j \in \mathbb{R}^m$ is the $j$-th vector of the word embedding matrix $\mathbf{E} \in \mathbb{R}^{|V| \times m}$ where $m$ is the dimension of embedding vectors, and $|V|$ is the dimension of vocabulary space. Let $\mathrm{Enc}(\cdot)$ and $\mathrm{Dec}(\cdot)$ represent the transformer encoder and decoder, respectively. For GLEN, we define the probability of generating an identifier $z$ from a document $d$ as follows.

$$P(z|d) = \prod_{t=1}^{n} P(z_t|d, \mathbf{d}_{<t}),$$
$$P(z_t = j|d, \mathbf{d}_{<t}) = \mathrm{Softmax}_j(\mathbf{d}_t \cdot \mathbf{E}^\top), \tag{3}$$

where $P(z_t = j|d, \mathbf{d}_{<t})$ denotes the probability that $z_t$ is the $j$-th token in the vocabulary space, and $\mathrm{Softmax}_j(\cdot)$ is the $j$-th element of the softmax function output. For the original transformer decoder, the output of each step, *i.e.*, $z_{<t}$, is fed for the next step. Here, we use the final hidden representations $\mathbf{d}_{<t}$ for the decoder input instead of $z_{<t}$. This method ensures that the decoder input does not fluctuate even if the document identifier fluctuates during training, allowing for stable training. (Empirically, we observed about 14.4% gain in Recall@1. See Section 5.2 for details.)

## 3.3 Two-phase Lexical Index Learning

To effectively train the lexical identifier, we introduce a two-phase training strategy: *keyword-based ID assignment* to learn the semantics of the corpus and the characteristics of identifiers and *ranking-based ID refinement* to adjust appropriate identifiers that encapsulate relevance signals.

### 3.3.1 Keyword-based ID Assignment

A document identifier should be concise yet informative, unlike a typical natural language sentence. Due to its unique nature, it is challenging to learn from scratch to assign appropriate identifiers to documents. We bridge the generation task gap between the pre-trained language model task and the identifier generation task by training the model to generate representative keywords. Specifically, we choose top-$n$ tokens with the highest tf-idf scores using BM25 (Robertson and Walker, 1994) as the keyword identifier $z^{\mathrm{key}}$ for the document. This ensures that the model can construct the semantics of the document from self-supervised signals extracted from the corpus and naturally learns the nature of identifiers. The model learns it using

sequence-to-sequence cross-entropy loss.

$$\mathcal{L}_{\mathrm{key}} = -\sum_{t=1}^{n} \log P(z_t^{\mathrm{key}}|d, \mathbf{d}_{<t}), \tag{4}$$

where $z_t^{\mathrm{key}}$ is a token at $t$-th step of $z^{\mathrm{key}}$. In addition, we also utilize a query as an input and train to infer keywords of its relevant documents.

### 3.3.2 Ranking-based ID Refinement

It is crucial to incorporate query-document relevance and learn how to generate identifiers of relevant documents from queries in training. To this end, we design two losses: (i) *pairwise ranking loss* for learning ranking and (ii) *pointwise retrieval loss* for learning the query-identifier relationship. Consequently, GLEN can dynamically learn how to generate lexical identifiers from the relevance signal.

**Pairwise ranking loss.** First, we introduce pairwise ranking loss, incorporating query-document relevance into identifier learning. It helps the model to represent queries as close to relevant documents $d^+$ and far from irrelevant documents $d^- \in \mathcal{N}$, where $\mathcal{N}$ is a set of negative documents obtained via prefix-aware dynamic negative sampling, which will be described later. The pairwise ranking loss is defined as follows.

$$\mathcal{L}_{\mathrm{pair}} = -\log \frac{\exp(\mathrm{rel}(q, d^+))}{\exp(\mathrm{rel}(q, d^+)) + \sum\limits_{d^- \in \mathcal{N}} \exp(\mathrm{rel}(q, d^-))}. \tag{5}$$

For pairwise ranking loss, we define the relevance score of a query $q$ and a document $d$ as described.

$$\mathrm{rel}(q, d) = \sum_{t=1}^{n} \mathbf{q}_t \cdot \mathbf{r}_t^\top,$$
$$\mathbf{r}_t = \mathrm{Softmax}(\mathbf{d}_t \cdot \mathbf{E}^\top / \tau) \cdot \mathbf{E}, \tag{6}$$

where $\mathbf{q}_t \in \mathbb{R}^m$ is the final hidden representation of the decoder for a query at time $t$. Note that $\mathbf{r}_t$ is used as a document representation, not $\mathbf{d}_t$. In the inference phase, the query should generate the identifier. In this regard, we exploit the identifier representation $\mathbf{r}_t$ for representing documents, thus mitigating the gap between training and inference. In addition, since $\arg\max(\cdot)$ to calculate $z_t$ in Eq. (2) is non-differentiable, we get $\mathbf{r}_t$ with $\mathrm{Softmax}(\cdot)$ and temperature $\tau$.

If $\tau$ is low enough, it yields a similar effect to $\arg\max(\cdot)$. However, when the model is not sufficiently trained, $\mathbf{r}_t$ may become collapsed regardless

of the document. As such, we adopt an annealed temperature, *i.e.*, $\tau = \max(10^{-5}, \exp(-t))$ where $t$ denotes the training epochs. (See Section 5.2 for the effectiveness of annealing).

**Pointwise retrieval loss.** To ensure the model can capture the relationship between the query and the relevant document identifier, we design a pointwise retrieval loss as follows:

$$
\begin{aligned}
\mathcal{L}_{\text{point}} = &- \sum_{t=1}^{n} \log P(z_t^+ | q, \mathbf{q}_{<t}) \\
&+ \lambda_{\text{dist}} \cdot \text{dist}(w^q, w^{d^+}), \\
w_t^q = &\ \mathbf{q}_t \cdot \mathbf{e}_{z_t}^\top \text{ for } t \in \{1, \dots, n\},
\end{aligned}
\tag{7}
$$

where $z^+$ indicates the identifier predicted from the positive document $d^+$, and $z_t^+$ is a $t$-th token of $z^+$. $\mathbf{e}_{z_t} \in \mathbb{R}^m$ is the word embedding vector of $z_t$. The first loss term is a cross-entropy loss that maps a query $q$ to the identifier $z^+$ of relevant documents. It alleviates the gap between training and inference in that mapping from queries to identifiers is performed in the inference. The second loss term utilizes the identifier logits $w^q$, $w^{d^+}$ and allows the model to learn the relative importance of the identifier tokens, *e.g.*, "Olympic" is more important than "list" in the example of Figure 1. Here, $\lambda_{\text{dist}}$ is the hyperparameter to adjust the importance between the pointwise loss terms. For distance function $\text{dist}(\cdot)$, we adopt cosine distance.

The final loss is the sum of the pairwise ranking loss and the pointwise retrieval loss:

$$
\mathcal{L} = \mathcal{L}_{\text{pair}} + \lambda_{\text{point}} \cdot \mathcal{L}_{\text{point}},
\tag{8}
$$

where $\lambda_{\text{point}}$ is the hyperparameter to control the importance of the pointwise retrieval loss. It enables end-to-end optimization of the retrieval task.

**Prefix-aware dynamic negative sampling**. To improve top-ranking retrieval performance robustly, we devise prefix-aware dynamic negative sampling for the pairwise ranking loss (Eq. (5)). As pointed out in Zhan et al. (2021), dynamic hard negatives, which are sampled during training based on retrieval results of the model itself, can effectively improve the ranking performance. To reflect the nature of the autoregressive model, we obtain a set of negative documents $\mathcal{N}$ based on the identifier prefix. Concretely, we determine the candidate negatives for each document in the following manner. Given an identifier length of $n$, we first take the documents that have the same identifier as the

target document, *i.e.*, we fetch documents with the same prefix for the first $n$ tokens. If the resulting set of documents does not reach the desired count of $N_{neg}$, we opt for documents with the same prefix at the first $n - 1$ tokens. We repeat it by reducing the length of the prefix until the set reaches $N_{neg}$ documents. Our approach iteratively samples documents based on the prefix. However, the cost was negligible in our experiments, and we found it effective for ranking, as shown in Section 5.2.

## 3.4 Collision-free Inference

The inference process of GLEN is straightforward: (i) we proceed over the documents for assigning identifiers to the document offline, and (ii) infer the identifiers of relevant documents from the query online. We finally assign a dynamically learned identifier to each document predicted by the model. We also employ constrained decoding to generate only the valid identifiers, and a ranked list of documents is obtained by beam search.

If an identifier is assigned to documents by learning, the documents with similar semantics may be mapped to the same identifier, *i.e.*, identifier collision. It incurs that documents with conflicting identifiers cannot be ranked. Existing studies (Wang et al., 2022; Tay et al., 2022) have appended additional digits (*e.g.*, X-X-0, X-X-1) to address this problem, but such manually defined identifiers may distort the subtle semantics of the identifier. On the other hand, we do not force the identifier to be unique for semantic learning of identifiers.

We introduce a novel solution, *collision ranking using identifier logit*, to resolve the collision issue at inference time. Specifically, we utilize a logit of each step in generating a lexical identifier $z$ from query $q$ (or document $d$). The relevance between a query and a document using identifier logits is defined as follows.

$$
\text{rel}_{ID}(q, d) = \cos(w^q, w^d).
\tag{9}
$$

For each query, we first rank the document identifiers via $P(z|q) = \prod_{t=1}^{n} (\mathbf{q}_t \cdot \mathbf{e}_{z_t}^\top / (\sum_i \mathbf{q}_t \cdot \mathbf{e}_i^\top))$. If multiple documents share a single identifier, they are ranked using $\text{rel}_{ID}(q, d)$. In this way, the collision problem can be avoided without unnecessary intervention in the semantic learning of identifiers. In particular, it has the advantage that there is only a negligible additional cost for ranking since the weights of the document identifiers $w^q, w^d$ are already used to compute $P(z|d), P(z|q)$.

## 4 Experimental Setup

### 4.1 Datasets

**Natural Questions (NQ320k)** (Kwiatkowski et al., 2019) consists of 320k query-document relevant pairs, 100k documents, and 7,830 test queries, which has been actively used in existing generative retrieval methods (Tay et al., 2022; Wang et al., 2022). We also follow the setup in Sun et al. (2023), splitting the test set into two subsets: *seen test* and *unseen test*. *seen test* consists of queries where the annotated target documents are included in the train set, while *unseen test* consists of queries where no labeled documents are included in the train set. **MS MARCO passage ranking (MS MARCO)** (Nguyen et al., 2016) is a large-scale benchmark dataset with 8.8M passages collected from Bing's results and 1M real-world queries. We use the official development set consisting of 6,980 queries with a full corpus, *i.e.*, 8.8M passages, following Ren et al. (2023). **BEIR** (Thakur et al., 2021) is a benchmark dataset for zero-shot evaluation on diverse text retrieval tasks. Following Sun et al. (2023), we assess on Arguana (Arg) (Wachsmuth et al., 2018) and NF-Corpus (NFC) (Boteva et al., 2016). For train data, we follow published train data constructed by Wang et al. (2022) for NQ320k for a fair comparison. For MS MARCO, we use the MS MARCO training set, which consists of 500k queries and randomly split 1,000 queries for validation. For the generated query of the MS MARCO, we used the published predicted queries [1].

### 4.2 Metrics

We report Recall and MRR for NQ320k following existing works (Sun et al., 2023). MRR@10 and nDCG@10 is the official metric of MS MARCO Passage Ranking and BEIR, respectively.

### 4.3 Baselines

We compare GLEN with three types of baseline models, including two sparse retrieval models (BM25 (Robertson and Walker, 1994) and DocT5Query (Nogueira and Lin, 2020)), four dense retrieval models (DPR (Karpukhin et al., 2020), ANCE (Xiong et al., 2021), Sentence-T5 (Ni et al., 2022a), and GTR (Ni et al., 2022b)), and six generative retrieval models. For generative retrieval methods, we categorize them following the

[1] https://github.com/castorini/docTTTTTquery

Table 1. (i) **Static numeric identifier.** DSI (Tay et al., 2022) uses a sequence-to-sequence model to generate numeric identifiers built by hierarchical k-means clustering. DSI-QG (Zhuang et al., 2023) and NCI (Wang et al., 2022) are built upon DSI while adopting augmented data via query generation and prefix-aware weight-adaptive decoder, respectively. (ii) **Static lexical identifier.** GENRE (Cao et al., 2021) utilizes a title as an identifier. SEAL (Bevilacqua et al., 2022) generates arbitrary n-grams to retrieve relevant documents, utilizing the FM-Index structure. TOME (Ren et al., 2023) performs retrieval by generating the document URLs via a two-stage generation architecture. (iii) **Dynamic numeric identifier.** GENRET (Sun et al., 2023) learns how to assign numeric identifiers based on a discrete auto-encoding scheme. (iv) **Dynamic lexical identifier.** To our knowledge, GLEN is the first work to employ the dynamic lexical identifier. For details of sparse and dense retrieval models, see Section A.1.

### 4.4 Implementation Details

We initialized GLEN with T5-base (Raffel et al., 2020). The batch size is set to 128, and the model is optimized for up to 3M steps and 30K steps using the Adam optimizer with learning rates 2e-4 and 5e-5 for keyword-based ID assignment and ranking-based ID refinement, respectively. We use beam search with constrained decoding and a beam size of 100 for inference. For two-phase lexical index learning, we set the length of document id $n = 3$, $n = 7$ after tuning among $\{2, 3, 5, 7, 10\}$ for NQ320k and MARCO, respectively. $\lambda_{\text{point}}$ and $\lambda_{\text{dist}}$ are set to 0.5 and 0.5 after tuning in $\{0, 0.25, 0.5, 1, 2, 4\}$, respectively. For $\tau$, we set it to 1e-5 with temperature annealing. We set the number of negative documents per query $N_{neg}$ to 8 after tuning in $\{0, 1, 2, 4, 8\}$ and adopted in-batch negatives, where all passages for other queries in the same batch are considered negative. Further details for model architecture and training hyperparameters can be found in Section A.1.

## 5 Experimental Results

### 5.1 Main Results

**Evaluation on NQ320k.** Table 2 presents the retrieval performance on the NQ320k. The key observations are as follows: (i) GLEN shows outperforming performance among sparse and dense baselines and competitive performance with generative re-

| Model | NQ320k (7,830) | | | Seen test (6,075) | | | Unseen test (1,755) | | |
|---|---|---|---|---|---|---|---|---|---|
| | R@1 | R@10 | MRR@100 | R@1 | R@10 | MRR@100 | R@1 | R@10 | MRR@100 |
| *Sparse & dense retrieval* | | | | | | | | | |
| BM25 (1994) | 29.7 | 60.3 | 40.2 | 29.1 | 59.8 | 39.5 | 32.3 | 61.9 | 42.7 |
| DocT5Query (2020) | 38.0 | 69.3 | 48.9 | 35.1 | 68.3 | 46.7 | 48.5 | 72.9 | 57.0 |
| DPR (2020) | 50.2 | 77.7 | 59.9 | 50.2 | 78.7 | 60.2 | 50.0 | 74.2 | 58.8 |
| ANCE (2021) | 50.2 | 78.5 | 60.2 | 49.7 | 79.2 | 60.1 | 52.0 | 75.9 | 60.5 |
| SentenceT5 (2022a) | 53.6 | 83.0 | 64.1 | 53.4 | 83.9 | 63.8 | 56.5 | 79.5 | 64.9 |
| GTR-base (2022b) | 56.0 | 84.4 | 66.2 | 54.4 | 84.7 | 65.3 | 61.9 | 83.2 | 69.6 |
| *Generative retrieval* | | | | | | | | | |
| GENRE (2021) | 55.2 | 67.3 | 59.9 | 69.5 | 83.7 | 75.0 | 6.0 | 10.4 | 7.8 |
| DSI (2022) | 55.2 | 67.4 | 59.6 | 69.7 | 83.6 | 74.7 | 1.3 | 7.2 | 3.5 |
| SEAL (2022) | 59.9 | 81.2 | 67.7 | - | - | - | - | - | - |
| DSI-QG (2023) | 63.1 | 80.7 | 69.5 | 68.0 | 85.0 | 74.3 | 45.9 | 65.8 | 52.8 |
| NCI (2022) | 66.4 | 85.7 | 73.6 | 69.8 | 88.5 | 76.8 | 54.5 | 75.9 | 62.4 |
| GENRET (2023) | 68.1 | **88.8** | **75.9** | 70.2 | **90.3** | 77.7 | **62.5** | **83.6** | **70.4** |
| TOME (2023) | 66.6 | - | - | - | - | - | - | - | - |
| **GLEN (Ours)** | **69.1** | 86.0 | 75.4 | **72.5** | 88.9 | **78.5** | 57.6 | 75.9 | 63.9 |

Table 2: Performance comparison for the proposed method and baseline models for NQ320k. The best generative retrieval model is marked **bold**, and the second best model is underlined. The number in parentheses indicates the number of queries. We refer to the results of baselines reported by Sun et al. (2023) and Ren et al. (2023). Results not available are denoted as '–'.

| Model | MS MARCO Dev (6,980) MRR@10 |
|---|---|
| *Sparse & dense retrieval* | |
| BM25 (1994) | 18.4 |
| DocT5Query (2020) | 27.2 |
| GTR-base (2022b) | 34.8 |
| *Generative retrieval* | |
| DSI (2022) | 3.1 |
| DSI-QG (2023) | 11.8 |
| NCI (2022) | 17.4 |
| **GLEN (Ours)** | **20.1** |

Table 3: Performance comparison for the proposed method and baseline models for MS MARCO passage dev. The best generative retrieval model is marked **bold**, and the second best model is underlined. We refer to the results of baselines reported by Pradeep et al. (2023).

| Model | BEIR (nDCG@10) | | |
|---|---|---|---|
| | Average | Arg | NFC |
| *Sparse retrieval* | | | |
| BM25 (1994) | 32.0 | 31.5 | 32.5 |
| DocT5Query (2020) | 33.9 | 34.9 | 32.8 |
| *Generative retrieval* | | | |
| DSI (2022) | 6.5 | 1.8 | 11.1 |
| NCI (2022) | 2.6 | 0.9 | 4.3 |
| GENRET (2023) | 12.1 | 12.1 | 12.1 |
| **GLEN (Ours)** | **16.8** | **17.6** | **15.9** |

Table 4: Zero-shot performance for the proposed method and baseline models for the BEIR dataset. The best generative retrieval model is marked **bold**, and the second best model is underlined. Average means the average accuracy over two datasets. We refer to the results of baselines reported by Sun et al. (2023).

trieval methods. Specifically, GLEN outperforms the best competitive generative retrieval model by +1.5% and +3.3% on Recall@1 in the NQ320k full and seen test set, respectively. (ii) While GLEN performs second best among generative models on unseen tests, it surpasses other models in seen tests, indicating that the ranking-based ID refinement effectively optimizes the identifiers. (iii) The generative retrieval methods with dynamic identifiers exhibit higher performance than those with static identifiers. It confirms their ability to capture the intricate semantics of documents and queries via end-to-end optimization.

**Evaluation on MS MARCO.** Table 3 shows the retrieval performance on the MS MARCO Passage Ranking set. GLEN yields a clear improvement over the best competitive generative retrieval methods and BM25 (Robertson and Walker, 1994) by 15.6% and 9.3% in MRR@10, respectively. Existing generative retrieval methods still struggle to memorize the knowledge of the corpus and thus often fail to work in large-scale corpora (Pradeep et al., 2023). In contrast, GLEN successfully works on large-scale corpora owing to learning identifiers via directly learning the relevance of queries and documents.

| Dataset (Metric) | Query subset (# queries) | Collision ranking | Random ranking |
|---|---|---|---|
| NQ320k (R@1) | All (7,830) | 69.1 | 68.6 |
| | Collision (441) | 41.5 | 31.9 |
| MARCO (MRR@10) | All (6,980) | 20.1 | 18.7 |
| | Collision (2,170) | 17.5 | 12.9 |

Table 5: Performance comparison of GLEN with different solutions for collision in inference. Random ranking denotes a randomly ranked result for colliding documents. We report an average of 10k runs.

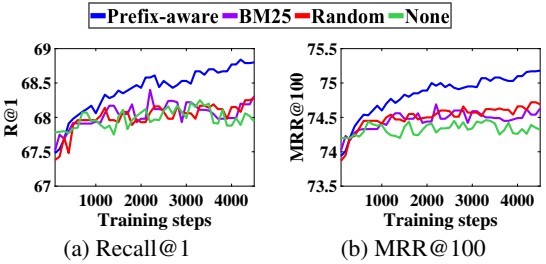

(a) Recall@1    (b) MRR@100

Figure 2: Performance comparison of GLEN depending on the negative sampling strategy by training step for ranking-based ID refinement phase on NQ320k.

**Zero-shot evaluation on BEIR.** We thoroughly investigate the generalization capability of GLEN via the zero-shot performance on the BEIR (Thakur et al., 2021) dataset after training on NQ320k, reported in Table 4. GLEN shows the best average accuracy in generative retrieval methods, surpassing the best competitive generative model by 38.8% on average. We also observe that the dynamic identifiers (*i.e.*, GLEN and GENRET) consistently outperform static identifiers, showing that they are more effective at capturing the complex semantics of documents and can be generalized in a zero-shot setting.

For dynamic identifiers, GLEN outperforms GENRET in a zero-shot evaluation. The difference in performance stems from two primary distinctions: (i) identifier types and (ii) solutions to identifier collisions. GLEN can assign a generalized identifier to a new document by leveraging knowledge from the PLM, while GENRET may have difficulty allocating numeric identifiers to new documents. Besides, the same identifier can be assigned to semantically similar documents, especially if the documents are out-of-domain. It often happens since models are not trained to differentiate between them. To rank these same identifier documents, GLEN introduces collision-free inference to break the tie, while GENRET cannot distinguish between them and places them randomly.

| Model | NQ320k (7,830) | | |
|---|---|---|---|
| | R@1 | R@10 | MRR@100 |
| **GLEN** | **69.1** | **86.0** | **75.4** |
| w/o keyword | 48.0 | 77.2 | 58.3 |
| w/o annealing | 67.6 | 85.2 | 74.2 |
| decoder input $z_{<t}$ | 60.4 | 83.7 | 69.1 |
| w/o pairwise loss | 67.3 | 84.7 | 73.8 |
| w/o pointwise loss | 68.3 | 85.8 | 74.8 |
| random negatives | 68.3 | 85.5 | 74.7 |

Table 6: Ablation study of GLEN. Note that keyword means keyword-based ID assignment, and annealing means temperature annealing for $\tau$ in Eq. (6).

## 5.2 In-depth Analysis

**Effect of collision-free inference.** As shown in Table 5, we validate the effectiveness of the collision ranking by comparing it with random ranking, which randomizes the ranking of colliding documents. For a thorough comparison, we further constructed a subset (*i.e.*, collision) by collecting queries where at least one labeled document is colliding documents. For NQ320k and MS MARCO datasets, we found that collision ranking improves performance by 30.2% and 35.5%, respectively, over random ranking for the collision query subset. MS MARCO dataset has a higher ratio of collision queries due to its larger corpora than NQ320k (*e.g.*, 8.8M vs. 109K). It verifies that colliding documents are effectively ranked without additional cost using identifier weight, while the semantics of identifiers are well preserved.

**Effect of prefix-aware dynamic negative.** Figure 2 depicts the effect of prefix-aware dynamic negative sampling along the training step. The prefix-aware dynamic negative exhibits the most effective performance for robust ranking, showing 1.0% and 1.2% gains in R@1 over the random negative and BM25 negative sampling, respectively. Furthermore, prefix-aware negatives delivered a 1.2% performance improvement in R@1, compared to not using hard negatives. It highlights that the nature of an autoregressive model is effectively reflected via the prefix, and dynamically sampled negatives are conducive to learning.

**Ablation study.** Table 6 presents the effectiveness of various strategies in GLEN on NQ320k. (i) The proposed keyword-based ID assignment phase remarkably improves accuracy by 44.0% in R@1. Notably, it successfully allows the model to learn the unique nature of identifiers based on self-supervised signals, thus enabling it to leverage

| Query | how would you represent g0 in the cell cycle of a neuron | | | |
|---|---|---|---|---|
| Relevance | Original title | GLEN ID | NCI ID | Keyword ID |
| ✓ | G0 phase | (#1) phase-phase-cell | (#14) 22-17-10-4 | phase-cells-nutri |
| ✗ | G2 phase | (#2) phase-phase-cell | (#2) 21-28-3-0 | phase-phase-cell |
| ✗ | Cell cycle checkpoint | (#3) point-check-cell | (#9) 1-27-21-1 | point-check-cell |
| ✗ | Neuron | (#57) neurons-nervous-neuro | (#1) 1-27-2-1 | neurons-nervous-neuro |
| ✗ | Neurotransmission | (#62) neuro-trans-apt | (#3) 1-27-2-2 | neuro-trans-apt |

Table 7: A retrieval example of GLEN and NCI on NQ320k. The number in parentheses denotes the rank of a document for each model. Keyword ID is the extracted identifier used at the keyword-based ID assignment phase of GLEN. Note that the tokenization process for GLEN ID and Keyword ID is simplified.

the knowledge of PLM. (ii) The temperature annealing for an identifier representation (in Eq. 6) contributes to stable training, yielding a 2.2% gain in R@1. (iii) Replacing decoder input $\mathbf{d}_{<t}$ to $z_{<t}$, the accuracy drops by 12.6%, suggesting that $\mathbf{d}_{<t}$ for decoder input enhances stable training. (iv) The proposed pairwise ranking loss and the pointwise retrieval loss contribute to the accuracy compared to adopting a single loss by up to 2.6% in R@1.

**Case study.** Table 7 exhibits a case study focusing on an identifier to elucidate how generative retrieval is performed. We take one query from NQ320k and show the retrieval results from GLEN and NCI. Our observations are as follows: (i) GLEN can assign the same identifier to documents with similar semantics (*e.g.*, "G0 phase" and "G2 phase"), but it effectively ranks them via collision-free inference. It indicates that GLEN successfully learns the subtle semantics of document-identifier relationships. (ii) GLEN refines identifiers through the refinement phase, changing from a keyword ID "phase-cells-nutri" to GLEN ID "phase-phase-cell". (iii) Static numeric identifiers in NCI fail to reflect the semantics of the documents. Although some documents are semantically similar (*e.g.*, "G0 phase" and "G2 phase"), they have completely different identifiers.

## 6   Conclusion

In this paper, we proposed a novel lexical index learning method, namely *Generative retrieval via LExical iNdex learning (GLEN)*. To effectively tackle the critical challenges of generative retrieval, we adopt a dynamic lexical identifier learning framework that can mitigate (i) the discrepancy between the knowledge of pre-trained language models and identifiers and (ii) the discrepancy between training and inference. GLEN successfully enjoys the benefits of a dynamic lexical document identifier via a delicately devised *two-phase index learning scheme* and *collision-free inference*. To

our knowledge, it is the first work introducing a dynamic lexical identifier for generative retrieval. Experimental results demonstrate that GLEN achieves state-of-the-art or competitive performance among generative retrieval methods.

## Limitations

This work proposes a new generative retrieval approach, GLEN, that dynamically learns lexical identifiers. Though we verified the performance of GLEN on a large corpus, it still exhibits a performance gap with longstanding conventional retrieval methods (*e.g.*, ColBERTv2 (Santhanam et al., 2022), LexMAE (Shen et al., 2023)), which still hold state-of-the-art performance. This implies that generative retrieval still faces limitations in learning large-scale corpus. It may require using large models or designing new training schemes, leaving many research problems to be explored. In addition, we experimentally verified the proposed model in a zero-shot setting. We showed that it outperforms the generative retrieval method but still performs less than sparse retrieval. It suggests that generative retrieval still suffers from limited generalization compared to well-designed dense or sparse retrieval models.

## Ethics Statement

This work complies with the ethics of ACL. The scientific artifacts we used are available for research with permissive licenses. The use of the artifacts in this paper adheres to their intended use.

## Acknowledgments

This work was supported by Institute of Information & communications Technology Planning & Evaluation (IITP) grant funded by the Korea government (MSIT) (No. 2019-0-00421, No. 2022-0-00680, and RS-2023-00219919).

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

# A Appendix

## A.1 Additional Experimental Setup

### A.1.1 Datasets

**Natural Questions (NQ320k)** (Kwiatkowski et al., 2019) is curated from Natural Questions. The dataset consists of a Wikipedia article and a query in the form of a natural language question. **MS MARCO passage ranking** (Nguyen et al., 2016) is a dataset released by Microsoft in 2016 for reading comprehension and adapted in 2018 for retrieval. For validation of MS MARCO, we randomly split 1,000 queries and use a corpus consisting of randomly sampled 100 passages from BM25 (Robertson and Walker, 1994) top 1000 passages for each query, *i.e.*, $|D| \approx 100{,}000$. **BEIR** (Thakur et al., 2021) is an evaluation benchmark dataset of 18 publicly available datasets from diverse text retrieval tasks and domains, which is widely used for evaluating the generalization capabilities of models. The task of Arguana and NFCorpus is argument retrieval and bio-medical information retrieval, respectively.

### A.1.2 Metrics

To measure the effectiveness, we use widely accepted metrics for information retrieval, including recall, mean reciprocal rank (MRR), and normalized discounted cumulative gain (nDCG), mean average precision (MAP) with retrieval size $K$. Recall is defined as $\frac{\sum_{i=1}^{N} rel_i}{k}$, where $i$ is the position in the list, $k$ is the number of relevant documents and $rel_i \in \{0, 1\}$ indicates whether the $i$-th document is relevant to the query or not. MRR is defined as $\frac{1}{|Q|} \sum_{i=1}^{|Q|} \frac{1}{rank_i}$, where $rank_i$ refers to the rank position of the first relevant document for the $i$-th query. nDCG considers the order of retrieved documents in the list. DCG@K is defined as $\sum_{i=1}^{K} \frac{2^{rel_i}-1}{log_2(i+1)}$ where $rel_i$ is the graded relevance of the result at position $i$. nDCG is the ratio of DCG to the maximum possible DCG for the query, which occurs when the retrieved documents are presented in decreasing order of relevance.

### A.1.3 Model architecture

GLEN is based on the transformer-based encoder-decoder architecture. The number of transformer layers is 12, the hidden size is 768, the feed-forward layer size is 3072, and the number of self-attention heads is 12 for the encoder and decoder. We implemented GLEN using PyTorch based on the Tevatron library [2] (Gao et al., 2023) and adopted the gradient cache (Gao et al., 2021) to accommodate large batch size with limited hardware memory.

### A.1.4 Baselines

BM25 (Robertson and Walker, 1994) is the traditional sparse retrieval model using lexical matching. DocT5Query (Nogueira and Lin, 2020) extends the document terms by generating relevant queries from the documents using T5 (Raffel et al., 2020). DPR (Karpukhin et al., 2020) is a bi-encoder model trained with in-batch negatives, which retrieves the documents via a nearest neighbor search. ANCE (Xiong et al., 2021) is a bi-encoder model trained with asynchronously selected hard training negatives. Sentence-T5 (Ni et al., 2022a) is similar

---

[2] http://tevatron.ai/

| Datasets | NQ320k | | | MS MARCO |
|---|---|---|---|---|
| Metrics | R@1 | R@10 | MRR@100 | MRR@10 |
| w/o refinement | 66.9 | 84.9 | 73.6 | 6.5 |
| w/ refinement | 69.1 | 86.0 | 75.4 | 20.1 |

Table 8: Ablation study of GLEN on the ranking-based ID refinement phase.

to DPR but utilizes T5 (Raffel et al., 2020) as a backbone. GTR (Ni et al., 2022b) is a scaled-up bi-encoder model with a fixed-size bottleneck layer based on Sentence-T5, which is a state-of-the-art dense retrieval model. For BM25, we followed the official guide[3] for reproducing. For NQ320k and BEIR, we refer to the results reported by Sun et al. (2023) and Ren et al. (2023). For MS MARCO, we refer to the results from Pradeep et al. (2023). The results of NCI are obtained based on the publicly released checkpoint by Wang et al. (2022).

### A.1.5 Reproducibility

The weight decay is 1e-4. We set the max sequence length for a query to 32, the max sequence length for a document to 156, and the dropout rate to 0.1. We conducted all experiments on a desktop with 4 NVidia RTX V100, 512 GB memory, and a single Intel Xeon Gold 6226.

### A.2 Effect of Ranking-based ID Refinement

Table 8 reports the effect of the ranking-based ID refinement phase of GLEN on NQ320k and MS MARCO. We observed that the refinement phase led to a performance gain of 3.3% for NQ320K and 209.4% for MS MARCO, respectively. This underscores the significance of the refinement phase, which trains on both pairwise ranking loss and pointwise retrieval loss as a key component in dynamic identifier learning. Also, it is shown that the ranking-based ID refinement phase is effective, especially for the large-corpus set (*i.e.*, MS MARCO). This is due to the fact that learning the mapping relations between documents and predefined identifiers becomes more challenging as the number of documents increases.

---

[3] http://pyserini.io/