# OpenReview forum: "GLEN: Generative Retrieval via Lexical Index Learning"
_EMNLP/2023/Conference — EMNLP 2023 Main_

### Official Review · Reviewer_L2RC · 2023-07-30

**Soundness:** 4

**Excitement:**

3: Ambivalent: It has merits (e.g., it reports state-of-the-art results, the idea is nice), but there are key weaknesses (e.g., it describes incremental work), and it can significantly benefit from another round of revision. However, I won't object to accepting it if my co-reviewers champion it.

**Paper Topic And Main Contributions:**

This paper proposes a new generative retrieval method that utilizes tokens as doc id representation instead of numbers or characters. The new id generation method bridges the gap between the knowledge of PLM and the identifier. Experimental result demonstrates its effectiveness.

**Questions For The Authors:**

Please refer to reasons to reject.

**Reasons To Accept:**

1. The new ID generation method seems novel and provides new insights for future research.
2. The writing is good and easy to follow and the experiments are extensive
3. The collision-free inference can well handle documents with conflicting identifiers without additional cost.

**Reasons To Reject:**

1. The new identifier generation is interpretable, however, the expressiveness of the identifier is doubtful, for example, 'olympic game list' can only represent part of the information hidden in document.
2. The experimental result in Table 2 cannot verify the effectiveness of this method considering the strong baseline GENRET.
3. Also, the experiment result in Tables 3 and 4 does not include the SoTA baseline like GENRET, which may not be fair.

**Reproducibility:**

3: Could reproduce the results with some difficulty. The settings of parameters are underspecified or subjectively determined; the training/evaluation data are not widely available.

**Reviewer Confidence:**

4: Quite sure. I tried to check the important points carefully. It's unlikely, though conceivable, that I missed something that should affect my ratings.

---

> ### Author Rebuttal · Authors · 2023-08-28
>
> Thank you for taking valuable time to review our paper. We deeply appreciate the comments on the novelty of the proposed method and its strengths in comparison to related works.  We address the three concerns mentioned in the 'Reasons To Reject' part.
>
> ### 1. The expressiveness of the identifier
>
> With the example identifier '*Olympic game list*,' it may seem insufficient to capture the meaning of the entire document. Nevertheless, during the ranking-based ID refinement phase, GLEN is trained to **generate the most appropriate identifier for query-document matching**. Therefore, we conjecture that GLEN encodes the document entirely and generates suitable identifiers that capture the core meaning of the document.
>
> In addition, to improve identifier expressiveness, **we can increase identifier length**. In our experiments, we observed a **25.0% performance gain** in MRR@10 for MS MARCO passage when we increased the identifier length from 3 to 10, as shown below.
>
>
>
> |           Model         |     MRR@10    |
> |-----------------------|-------------|
> |           BM25          |      18.4     |
> |        DocT5Query       |      27.2     |
> |         GTR-base        |      34.8     |
> |            DSI          |       3.1     |
> |          DSI-QG         |      11.8     |
> |            NCI          |      17.4     |
> |      GLEN (length=3)    |      18.0     |
> |     GLEN (length=10)    |      22.5     |
>
>
> **GLEN (length=10) outperforms BM25** and the generative retrieval baselines. However, as the identifier length increases, decoding time also increases, which can lead to a trade-off between expressiveness and efficiency. (For MS MARCO, GLEN (length=10) takes approximately 2.27 times longer than GLEN (length=3) for retrieval.) Therefore, choosing the appropriate identifier length is nontrivial to balancing expressiveness and efficiency. We have chosen a length of 3.
>
>
>
> ### 2. Effectiveness of GLEN in NQ320k (Table 2)
>
> From Tables 2 and 4, we can see the performance difference between GENRET and GLEN **depending on the dataset**, i.e., NQ320k and BEIR. That is, the performance difference between GLEN and GENRET is not significant for the trained dataset, NQ320k, but it seems to be significant for the out-of-domain dataset, BEIR. Based on these results, we conjecture that GLEN performs better in more general situations, although both models showed similar accuracy on the NQ320k with a small corpus size.
>
> We hypothesize that the performance differences across datasets come from the following reasons. First is the **generalization capability of lexical identifiers**. GLEN can assign a generalized identifier to documents by leveraging knowledge from the PLM, whereas GENRET may struggle with it. This is crucial when processing new documents or when the corpus size is large. Second is the **collision-free inference** of GLEN. If multiple documents have the same identifier, GLEN breaks the tie using the logit of the identifier, while GENRET ranks them in random order. This collision issue can become more prevalent with out-of-domain corpus sets or larger corpus sizes and must be handled.
>
> Consequently, we believe that the effectiveness of GLEN is verified, especially in the zero-shot setting.
>
> ### 3. Comparison of GLEN and GENRET for MS MARCO (Table 3) and BEIR (Table 4)
>
> #### (1) MS MARCO passage dataset
>
> **We strongly agree that an experimental comparison between GLEN and GENRET** on the MS MARCO passage is needed, and we are working on it.
>
> Since the results of GENRET on MS MARCO passage full corpus set are not publicly available, **we have been trying to reproduce GENRET for a long time**. First, we requested the code for the authors and received a response that they would publish it on Github later. However, as of now, there is no code yet. (https://github.com/sunnweiwei/GenRet)
>
> Therefore, we have tried to implement GENRET but failed to replicate the experimental results of the paper. Specifically, we observed that our GENRET implementation generates the same IDs for all documents after training. When we contacted the author about this issue, the author told us that the distribution of docid values would be highly uneven without codebook initialization and docid re-assignment technique, both of which were introduced in the GENRET paper for docid diversity. We are trying to solve the issue by focusing on the two techniques. **After reproducing GENRET, we will compare the results in detail with GLEN**. We sincerely understand your concern about the lack of comparison to GENRET in Table 3 and will do our best to address it.
>
> #### (2) BEIR dataset
> Table 4 shows the comparison with GENRET on the BEIR dataset, and the experimental results show that **GLEN has a higher generalization capability than GENRET**. For further details, please refer to Table 4. We will also compare performance on other retrieval tasks in BEIR after reproducing GENRET.

---

### Official Review · Reviewer_XCay · 2023-07-31

**Soundness:** 5

**Excitement:**

4: Strong: This paper deepens the understanding of some phenomenon or lowers the barriers to an existing research direction.

**Paper Topic And Main Contributions:**

This work proposes GLEN, a generative retrieval model based on lexical document identifier. The proposed model is trained with the objectives to both align language model vocabulary and semantic representations. To deal with lexical document ID conflicts, the model uses a retrieval model to select best candidate using weight matches.

The main contribution of this work, in my opinion, is showing generative retrieval model is able to achieve top effectiveness (especially in zero-shot and cross-domain retrieval) while taking a different approach from the previous best generative retrieval models using numeric identifiers.

**Reasons To Accept:**

This work is simple but potentially impactful. The methodology is innovative as it is the first to explore a new dimension, i.e., Lexical dynamic generative retrieval. The methodology is also well depicted and structured, with clear formula and math notations to assist understanding.

The obtained results look very promising, especially in zero-shot / cross-domain effectiveness (according to Table 4). As generative retrieval is a newly explored retrieval model, I look forward to more development in this direction.

Importantly, the authors claim they are willing to share source code upon acceptance, this would be convenient for other researchers to replicate and better inspect this work.

**Reasons To Reject:**

* Since the dynamic learning is a critical dimension to differentiate this model from others (in terms of novelty), it should be emphasized the side effects of adapting dynamic hard negatives. Does it require re-indexing documents compared to non-generative retrieval?

* Although being similar to the indexing model, the implantation of retrieval model is missing. Is it initialized with the same weight as the indexing model? How about using only the indexing model? (e.g., using the indexing model of the last training time step as retrieval model)

* Table 6 is informative, but it would be great to perform significance test for reader to better gauge the importance of each component.

**Reproducibility:**

5: Could easily reproduce the results.

**Reviewer Confidence:**

4: Quite sure. I tried to check the important points carefully. It's unlikely, though conceivable, that I missed something that should affect my ratings.

---

> ### Author Rebuttal · Authors · 2023-08-28
>
> We truly appreciate your constructive comments. Also, thank you for considering our work as a potentially impactful, innovative, and promising methodology. We address the points you mentioned in the ‘Reasons To Reject’ part.
>
> ***
>
> ###  1. Does it require re-indexing documents compared to non-generative retrieval?
>
> **No**, GLEN's dynamic identifier learning does not require re-indexing documents compared to non-generative retrieval. The dynamic learning is an essential point that distinguishes GLEN from other generative models, but the **inference process is still in the nature of generative retrieval**. In other words, instead of building an inverted index or storing dense vectors in non-generative retrieval, GLEN only requires the document identifier for each document to perform a search.
>
> ### 2. Side effects of dynamic hard negative
>
> Compared to static hard negative sampling (e.g., BM25 hard negatives), **our dynamic hard negative sampling can be more costly in terms of time efficiency**. For each document, static hard negatives can be sampled from a predefined negative document set beforehand. Therefore, sampling during training time takes very little time. On the other hand, our dynamic hard negatives require additional time to sample documents with the same prefix as the positive document. (In our experiments, this was negligible.) We appreciate your feedback, and we will revise the paper to clarify this point.
>
> ### 3. Implementation of indexing model and retrieval model
>
> **The two models have the same architecture**, and as mentioned on line 232, they also **share parameters**.  We will modify this to enhance clarity and avoid any confusion.
>
> ### 4. Significance test for Table 6 (ablation study)
>
> We thank the reviewer for pointing this out. We conducted Bonferroni correction pairwise t-tests following [2,3]. Statistical significant differences (p<0.01) with Bonferroni correction between the GLEN and variants are denoted as ∗. **We will add the results** in the paper.
>
> |             Model            |       Recall@1       | Recall@10 | MRR@100 |
> |----------------------------|---------------|----|-------|
> |             GLEN             |       68.3      | 85.7 |   74.9  |
> |       w/o keyword-based      |       46.9*     | 76.5*|   57.5 *|
> |         w/o annealing        |       67.5      | 85.2 |   74.1  |
> |  w/o adaptive decoder input  |       60.4*     | 83.7 |   69.1 *|
> |   w/o pairwise ranking loss  |       67.3      | 83.8 |   73.4  |
> | w/o pointwise retrieval loss |       67.0      | 85.8 |   74.1  |
> |          random neg          |       67.9      | 85.1 |   74.3  |
>
>
>
> ### References
> - [1] Vladimir Karpukhin et al., Dense Passage Retrieval for Open-Domain Question Answering. EMNLP 2020
> - [2] Norbert Fuhr, Some Common Mistakes In IR Evaluation, And How They Can Be Avoided. SIGIR Forum 51(3): 32-41 (2017)
> - [3] Leonid Boytsov et al., Deciding on an adjustment for multiplicity in IR experiments. SIGIR 2013

---

### Official Review · Reviewer_CqNg · 2023-08-04

**Soundness:** 3

**Excitement:**

4: Strong: This paper deepens the understanding of some phenomenon or lowers the barriers to an existing research direction.

**Paper Topic And Main Contributions:**

This paper discusses a generative retrieval approach that uses a learned, dynamic lexical identifier for the retrieval candidates. The novelty of the work comes from the learning process of the dynamic lexical identifier and how this learning process interacts with the ranking mechanism. This is certainly an interesting work that deserves a spot at the conference. However, it has some drawbacks that would definitely benefit from some revision.

1. The closest prior work is probably GENRET, which learns a numeric document identifier instead of a lexical one. The authors also acknowledge the similarity but did not provide more comparisons and analyses between them. In the result section, the authors generally conclude that both the proposed GLEN and GENRET outperform models using static identifiers, which is a fine conclusion(e.g., Line 485 and 508). However, the novelty and the motivation of the work come from the leverage of using lexical identifiers to better utilize the language model. The difference is certainly small in NQ320k but much larger in BEIR. It would be much more complete if the authors provided more analysis into the reasons why we are seeing such discrepancies in the two collections.
2. The authors added additional steps, such as pairwise and pointwise ranking losses, to enhance the ranking ability of the model. These are more independent of the central idea proposed in the paper, which makes it hard to compare with other works. Since the authors did not reimplement other models (like a lot of other papers), it is likely to be something that the authors cannot fix. However, this raises the question of what if we also apply these training techniques to other models.
3. The collision-resolution approach is an interesting one. If I understand this correctly, this is basically falling back to a dense retrieval approach that uses the intermediate representation to break the tie (due to the identifier being the same). It would be interesting to see if we use a simple BM25 or other dense retrieval models to break the ties, which is much more practical, instead of random ranking in Table 7.

In summary, while there is room for this paper to improve, I believe these are generally fixable. The paper, as it is, has already provided a good set of evidence for the effectiveness and novelty of the work. Therefore, I vote for acceptance for this paper.

**Questions For The Authors:**

- If we already need to have a two-step process for learning/assigning document identifiers and learning the relevancy, this is already very similar to any other dense retrieval model, which first trains the model and then indexes the documents. What is the advantage and motivation of using a generative retrieval model?
- Have you looked into the version of not learning the document identifier but directly assigning them using tokens with high tfidf scores (like the supervision of the ID training step)?

**Reasons To Accept:**

Provide a novel approach for determining the document identifier in the generative retrieval model.

**Reasons To Reject:**

Lack of detailed analysis to its dynamically-generated numeric identifier counterpart (GENRET).

**Reproducibility:**

4: Could mostly reproduce the results, but there may be some variation because of sample variance or minor variations in their interpretation of the protocol or method.

**Reviewer Confidence:**

4: Quite sure. I tried to check the important points carefully. It's unlikely, though conceivable, that I missed something that should affect my ratings.

**Typos Grammar Style And Presentation Improvements:**

- The word “identifier” is overloaded in the paper. It can mean “predicted,” “generated,” or just generic IDs throughout the paper. It would be better to specify what exactly it should refer to whenever you mention it.
- There are quite a lot of notations in the paper; some are similar to each other but with a different symbol because of the version or usage. It would be nice to explain the naming rule if there is one so it is easier for the readers to follow.
- It would be nice to include ColBERT in the comparison since it is generally the state-of-the-art retrieval architecture regarding effectiveness.
- Some papers cited are already accepted in some conferences. It is a better practice to cite their published/latest version instead of the preprint one.

---

> ### Author Rebuttal · Authors · 2023-08-28
>
> We sincerely appreciate your valuable comments that will definitely improve our paper. Also, thank you for acknowledging the novelty of our work. First, we would like to address three points mentioned in the ‘Paper Topic And Main Contributions’ part.
>
> ***
> ### 1. Comparison and analysis between GLEN and GENRET [1]
>
> Thanks to the reviewer for pointing it out. We will include the following analysis to provide
> clear motivation for our work. We have analyzed why the performance difference between GENRET and GLEN depends on the dataset, i.e., NQ320k and BEIR. The **main differences between GLEN and GENRET** can be summarized as follows.
>
> |  Model  | Identifier type | Identifier collision solution          |
> |--------|-----------------|----------------------------------------|
> | GLEN   | lexical         | Collision-free ranking using ID weight |
> | GENRET | numeric         | Random ranking                         |
>
> Based on this analysis, we hypothesize that the discrepancy is due to the following reasons: **(1) the generalization capability of lexical identifiers**, and **(2) the collision-free inference of GLEN for new documents**.
>
> First, lexical identifiers are likely to have better generalization capability in a zero-shot setting than numeric identifiers. The key to generalization is to assign appropriate identifiers to documents that represent out-of-domain documents well. **GLEN can assign a generalized identifier to a new document by leveraging knowledge from the PLM, whereas GENRET may struggle with it**. For instance, if a new document belongs to the medical domain and the training set lacks that domain, GLEN can assign domain-specific lexical terms as identifiers. This is because GLEN’s lexical identifier units are pre-trained T5 vocabulary tokens, while GENRET’s numeric identifier units are numbers and are only trained to represent training documents. Therefore, the performance difference between GLEN and GENRET is not significant for the trained dataset, NQ320k. In contrast, it seems significant for the untrained dataset, BEIR.
>
> Second, the collision-free inference of GLEN would work better in BEIR, which performs retrieval for new documents. If multiple documents have the same identifier, **GLEN breaks the tie using the logit of the identifier, while GENRET ranks them in random order**. (Refer to Section 5.2 of [1]) For in-domain documents, the model can be trained to distinguish between similar documents, but for out-of-domain documents, the same identifier can be given if they are semantically similar. GLEN and GENRET differ in their collision handling, hence the performance gap seems to be relatively larger for BEIR than NQ.
>
>
>
>
> ### 2. Role of ranking-based ID refinement phase (i.e., pairwise ranking loss and pointwise retrieval loss)
>
> First, we would like to clarify that the **ranking-based ID refinement phase is very crucial** to our main idea. The pairwise ranking loss and pointwise retrieval loss are key to dynamic identifier learning, which is our main idea.
>
> Suppose there is only a keyword-based ID assignment phase. In that case, the model will only learn to map documents or queries to predefined identifiers, which is highly similar to the learning process of other models that utilize static identifiers (e.g., NCI [2]). We believe that the ranking-based ID refinement phase is essential to learn appropriate identifiers for query-document matching. It is shown by the following **experimental results** as follows.
>
>
> | NQ320k           | Recall@1 | Recall@10  |  MRR@100 |
> |--------------|----------|------------|---------|
> | w/o refinement phase    |   66.8 |     84.3 | 73.3 |
> | w/ refinement phase   |   68.2 |     85.9 |   74.8 |
>
> | MS MARCO | MRR@10     |
> |-------------|------------|
> |  w/o refinement phase  |      4.0 |
> |  w/ refinement phase | 18.0 |
>
> Also, it is shown that the **ranking-based ID refinement phase is effective, especially for the large-corpus set (e.g., MS MARCO)**. This is because learning the mapping relations between documents and predefined identifiers becomes more difficult as the number of documents increases.
>
> We'd like to thank you for the suggestion. It is a very interesting idea to apply our ranking-based ID refinement to other models, such as NCI. We are in the process of implementation, and we believe it might improve the accuracy of those models as well.
>
>
>
> ### 3. Extension of collision-free approach using other retrieval models
>
> We appreciate your suggestion and have observed very interesting results. In alignment with your proposal, we have investigated the **utilization of other retrieval models to effectively deal with collisions**. We present the results obtained using BM25 model below.
>
>
> | NQ320k           | Recall@1 | Recall@10  | MRR@100 |
> |----------------|----------|------------|---------|
> | random rerank  |   67.7 |     82.4 |  74.5 |
> | BM25 rerank    |   68.9 |     86.0 |  75.3 |
>
> | MS MARCO |  MRR@10     |
> |----------------|------------|
> | random rerank  |       16.6 |
> | BM25 rerank    |      19.2 |
>
> The experimental results demonstrate that BM25 reranking provides a 1.8% gain in NQ (R@1) and a 15.6% gain in MARCO (MRR@10) compared to random reranking.
> We believe that this is a very simple yet effective way to improve the performance of the model, and it is expected that utilizing other dense retrieval models will further improve the effectiveness. However, unlike the proposed collision-free inference, which requires little additional computational cost or storage overhead, **utilizing an additional retrieval model has the limitations of (1) additional retrieval latency and (2) the need for an auxiliary index structure**. Therefore, we adopt an identifier weight-based reranking that eliminates the need for additional cost.
>
> ***
> Next, we would like to answer the ‘Questions For The Authors’ part in the following.
>
>
> ### 4. What is the advantage and motivation of using a generative retrieval model?
>
> GLEN is similar to dense retrieval in that it learns relevance signals between queries and documents through ranking loss during training. However, GLEN is only trained with additional losses, and the inference phase is identical to existing generative retrieval methods. That is, GLEN, like generative retrieval models, does not require an auxiliary index structure for inference. In this regard, the advantages of generative retrieval are as follows.
>
> **(1) Efficiency in terms of memory storage**. The memory footprint of a generative retrieval model depends solely on the model parameters. This contrasts with dense or sparse retrieval models, which require additional storage space to store documents depending on document collection size (e.g., the inverted index or ANN index).
>
> **(2) Efficiency in terms of retrieval latency**. For online retrieval latency, the generative retrieval model only requires the time to generate an identifier for a query. According to [1], for the representative models, GTR-base [3] and GENRET, the online retrieval latency is 1.97 seconds and 0.16 seconds (beam size=100), respectively, but the performance is similar at 57.6 and 58.1 based on MRR@10.
>
> **(3) Effectiveness via end-to-end optimization**. Generative retrieval can optimize the entire retrieval process in an end-to-end manner, thus enabling effective retrieval.
>
> Despite these advantages, the generative retrieval model still has a performance gap with existing state-of-the-art conventional retrieval methods. In this sense, there is still much room for further research. Furthermore, we will enhance the clarity of our paper by providing a more explicit comparison between dense retrieval and GLEN.
>
> ### 5. Have you looked into the version of not learning the document identifier but directly assigning them using tokens with high TF-IDF scores (like the supervision of the ID training step)?
>
> We appreciate your interesting suggestion. Here is the experimental result when the document identifier is not dynamically learned, but a predefined keyword-based identifier is assigned based on a high TF-IDF score. Note that for predefined identifiers, we cannot utilize the proposed collision-free inference since the logit of the identifier does not exist. So, for a fair comparison, we report the results of random reranking for both pre-defined identifiers and learned identifiers.
>
> | NQ320k   | Recall@1 | Recall@10  | MRR@100 |
> |--------------|----------|------------|---------|
> | GLEN with pre-defined identifier | 67.6 | 85.9 |  74.4 |
> | GLEN with learned identifier | 67.7 |     82.4 |   74.5 |
>
> | MS MARCO |  MRR@10     |
> |-------------|------------|
> | GLEN with pre-defined identifier |  6.8 |
> | GLEN with learned identifier |      16.6 |
>
>
> Experimental results show that utilizing learned identifiers improves performance over predefined identifiers for NQ and MS MARCO. In particular, in large-scale corpus like MS MARCO, it is more effective to assign identifiers through learning rather than predefining them. We conjecture that learning the mapping relationship between documents and predefined identifiers becomes more difficult as the number of documents increases.
>
>
> ***
> Here is our response to ‘Typos Grammar Style And Presentation Improvements.’
>
> ### 6. Comparison with other SOTA dense retrieval models (e.g., ColBERT)
>
> Your constructive suggestions are consistent with our views. We also believe that it is necessary to compare the performance with state-of-the-art sparse or dense retrieval models such as ColBERT in addition to GTR-base [3]. We are currently implementing ColBERT on the same code framework as GLEN. We will compare our experimental results with GLEN in the future for completeness.
>
>
>
> ### 7. Clarifying descriptions and correcting citations
>
> We appreciate your detailed comments about our paper. (1) In terms of the word 'identifier'. We would like to revise the description of predicted, generated, and generic identifiers to match their meaning. (2) For the naming rule of notation, we basically use bold for vectors and italics for scalars and text sequences. We will add additional explanations to the text to make it clearer. (3) We will modify citations to follow published versions for [4, 5, 6, 7].
>
> ### References
> - [1] Weiwei Sun et al., Learning to Tokenize for Generative Retrieval. CoRR abs/2304.04171 (2023)
> - [2] Yujing Wang et al., A Neural Corpus Indexer for Document Retrieval. NeurIPS 2022
> - [3] Jianmo Ni et al., Large dual encoders are generalizable retrievers. EMNLP 2022
> - [4] Luyu Gao et al., Tevatron: An efficient and flexible toolkit for neural retrieval. SIGIR 2023
> - [5] Hyunji Lee et al., Nonparametric decoding for generative retrieval. ACL (Findings) 2023
> - [6] Ruiyang Ren et al., TOME: A two-stage approach for model-based retrieval. ACL 2023
> - [7] Shengyao Zhuang et al., Bridging the Gap Between Indexing and Retrieval for Differentiable Search Index with Query Generation. Gen-IR@SIGIR 2023

---

### Meta-Review · Area_Chair_h6S2 · 2023-09-24

**Recommendation:** 5

**Metareview:**

This paper discusses a generative retrieval approach that uses a learned, dynamic lexical identifier for the retrieval candidates. The proposed model, named Generative retrieval via LExical iNdex learning (GLENT), exploits a dynamic lexical identifier using a two-phase index learning strategy. The work is novel and interesting. It had some drawbacks, that were clearly identified by the reviewers and that were well addressed in the rebuttal.

---

### Decision · Program_Chairs · 2023-10-07

**Decision:**

Accept-Main

**Comment:**

This paper discusses a generative retrieval approach that uses a learned, dynamic lexical identifier for the retrieval candidates. The proposed model, named Generative retrieval via LExical iNdex learning (GLENT), exploits a dynamic lexical identifier using a two-phase index learning strategy. The work is novel and interesting. It had some drawbacks, that were clearly identified by the reviewers and that were well addressed in the rebuttal.